# A Tale of Two Proteases: M^Pro^ and TMPRSS2 as Targets for COVID-19 Therapies

**DOI:** 10.3390/ph16060834

**Published:** 2023-06-02

**Authors:** Barbara Farkaš, Marco Minneci, Matas Misevicius, Isabel Rozas

**Affiliations:** School of Chemistry, Trinity Biomedical Sciences Institute, Trinity College Dublin, 152-160 Pearse Street, D02 R590 Dublin, Ireland; barbara.farkas95@gmail.com (B.F.); minnecim@tcd.ie (M.M.); misevicm@tcd.ie (M.M.)

**Keywords:** SARS-CoV-2, MPro, TMPRSS2, antiviral, protease inhibitor

## Abstract

Considering the importance of the 2019 outbreak of severe acute respiratory syndrome coronavirus 2 (SARS-CoV-2) resulting in the coronavirus disease 2019 (COVID-19) pandemic, an overview of two proteases that play an important role in the infection by SARS-CoV-2, the main protease of SARS-CoV-2 (M^Pro^) and the host transmembrane protease serine 2 (TMPRSS2), is presented in this review. After summarising the viral replication cycle to identify the relevance of these proteases, the therapeutic agents already approved are presented. Then, this review discusses some of the most recently reported inhibitors first for the viral M^Pro^ and next for the host TMPRSS2 explaining the mechanism of action of each protease. Afterward, some computational approaches to design novel M^Pro^ and TMPRSS2 inhibitors are presented, also describing the corresponding crystallographic structures reported so far. Finally, a brief discussion on a few reports found some dual-action inhibitors for both proteases is given. This review provides an overview of two proteases of different origins (viral and human host) that have become important targets for the development of antiviral agents to treat COVID-19.

## 1. Introduction

The outbreak of the pathogenic severe acute respiratory syndrome coronavirus 2 (SARS-CoV-2) in 2019 and the rapid spread of the resulting coronavirus disease 2019 (COVID-19) pandemic signified probably the most serious global health emergency in the last decades. That said, the emergence of beta-coronaviruses started in the new millennium with two of the largest pathogenic pandemic outbreaks since the Spanish flu (1918–1920) as follows [1,2,3,4]: first, in 2002, the severe acute respiratory syndrome coronavirus (SARS-CoV) infected over 8000 people with a 10% mortality; second, in 2012, the mortality figure increased when Middle East respiratory syndrome coronavirus (MERS-CoV) caused 35% causalities in 2300 cases [5]. The emergence of SARS-CoV-2 in 2019 has seen all of the two decades worth of research efforts to put beta-coronaviruses to test in a race of drug development versus continuous increase in the virus transmissibility and mutation [6]. COVID-19’s devastating effects rapidly reached an infection rate of over 10% of the human population and showed no discrimination in geographical regions [7]. Currently, COVID-19 has resulted in more than 6.9 million deaths according to the World Health Organization (WHO) [8]; however, despite the enormity of this global health crisis, no small molecule or peptidomimetic therapy has been found. As the search continues and the ‘weak’ points of the SARS-CoV-2 viral genome evolve, a more strategic approach of dual targeting might be needed to combat this major pathogen.

SARS-CoV-2 belongs to the coronavirus family, more precisely to the beta-coronavirus 2B lineage encompassing RNA viruses with the longest genomes of all known single-stranded RNA viruses, whose cooperation between the RNA polymerases, RNA helicases, and proofreading exonucleases grants efficient transcription of their lengthy genome among both humans and animals [9]. Complete genome sequences resolved in the early stages of the pandemic were shown to share 79.5% of the sequence identity with the 2002 SARS-CoV virus. The correlation search was expanded to other beta-coronaviruses with a striking 96% match with the bat coronavirus strain BatCov RaTG13, indicating a potential origin and structural evolution path of SARS-CoV-2 [10].

All beta-coronaviruses contain a remarkably large RNA genome (approximately 29.9 kb) and distinctive spikes arising from their surface giving them the appearance of a solar corona (Figure 1). The SARS-CoV-2 virus carries four structural proteins, nucleocapsid proteins (N), membrane protein (M), spike protein (S), and envelope protein (E) as well as sixteen non-structural proteins (nsp1-16). The N proteins form the capsid surrounding the genome, which is further covered by an envelope associated with M, S, and E structural proteins [11]. The non-structural proteins have different functions related to replication and transcription, processing of the polyprotein during replication, modulation of the survival signalling pathway of the host cell, modification of the membrane of the endoplasmic reticulum (ER), generation of replication organelles, combining the template-primer RNA, binding ssRNA, and capping the methylation of viral mRNAs, or deoxyribonuclease activity, all of them aiming to inhibit the defenses of the host cells [12].

The replication cycle of SARS-CoV-2 starts with the attachment of the virus into the human host cell by the binding of the S protein of the viral spikes to the angiotensin-converting enzyme 2 (ACE2) receptor, which is expressed on the surface of cells in the lungs, arteries, heart, kidneys, and intestines [13]. The S protein has two subunits, the surface unit S1, involved in the attachment, and the unit S2 involved in the fusion of viral and host membranes (Figure 1). After the attachment to ACE2, the host transmembrane protease serine 2 (TMPRSS2) cleaves the viral S protein (more details in Section 2.2.2) promoting fusion at the cellular membrane and allowing the entry of the virus in the host to release the (+) ssRNA genome [14]. Next, viral translation takes place, producing two large polyproteins (i.e., pp1a and pp1ab) that require cleavage from different viral proteases, the papain-like protease (PL^Pro^) and the chymotrypsin-like or main protease (3CL^Pro^ or, more recently and in the rest of this review M^Pro^), to form new structural proteins and enzymes [15]. Subsequently, the new proteins together with the RNA create the replication complex that serves as a template to transcribe and replicate RNA, thus amplifying the viral (+)ssRNA. Finally, the newly synthesized viral proteins and (+) ssRNA copies assemble to form the new virions [16], which mature and are released from the host cell by exocytosis, repeating the infection cycle several times [17].

Given the severity of the pandemic, drug repurposing was extensively pursued as a shortcut to find therapies for COVID-19. To date, only three antiviral small-molecule drugs (i.e., remdesivir, molnupiravir, and nirmatrelvir, Figure 2, Table 1) have been commercialised as COVID-19 treatments, and one (i.e., emsiltrevir, Figure 2, Table 1) has obtained emergency approval; most of these four antivirals were already investigative drugs being developed for other viral infections [18,19,20,21].

The first compound approved to treat COVID-19 was remdesivir (developed by Gilead, see Figure 2, upper left, Table 1), which was originally proposed for the treatment of hepatitis C and subsequently investigated for Ebola virus infection before being studied as a post-infection treatment for COVID-19 [19]. This is a ProTide type of prodrug, in which triphosphate metabolite inhibits RNA polymerase, and it is intravenously administered [22]. In the EU, it is indicated for the treatment of COVID-19 in adults and adolescents with pneumonia requiring supplemental oxygen [23]. In the US, it is suggested to be used in adults and adolescents for the treatment of COVID-19 requiring hospitalization [24]. The second drug put in the market as a COVID-19 therapy was molnupiravir (developed by Merck, see Figure 2, upper centre, Table 1), originally developed against the influenza virus and supposedly abandoned for mutagenicity problems [25,26]. After further evaluation and studies, it was found to be active against SARS-CoV-2 by exerting its antiviral action through the introduction of copying errors during viral RNA replication. This drug is administered orally, and even though the company claimed a decrease in the risk of hospitalization or death from COVID-19 by 50% when they filed for emergency use authorization, this claim was reduced to 30% when finally approved by the FDA [27]. The most recent therapy approved as an anti-COVID-19 agent has been paxlovid (developed by Pfizer, see Figure 2, down, Table 1), which consists of two components, an experimental drug called nirmatrelvir and a known drug ritonavir used for HIV infections [28]. Both components are viral protease inhibitors, meaning they block an enzyme that divides newly translated long viral polyproteins into smaller, functional, or structural proteins. While ritonavir is an inhibitor of the HIV protease, nirmatrelvir inhibits M^Pro^ [21]. Early work on nirmatrelvir showed its application against human rhinoviruses, and further development prior to the COVID-19 pandemic showed applicability against other viruses including SARS-CoV [29]. Paxlovid was approved by the FDA for emergency use in December 2021, and it was later approved in the UK, EU, and Canada in the following months [30,31,32,33]. This drug is administered orally and according to Pfizer, it reduces the risk of hospitalization or death by 88% [34]. Finally, ensitrelvir (developed by Shionogi, Figure 2, upper right, Table 1), which is an anti-COVID-19 agent in Phase III, obtained emergency approval in November 2022 in Japan [35]. This oral non-peptidic drug acts as a non-covalent inhibitor of the M^Pro^ in SARS-CoV-2, and it was discovered via virtual screening of an in-house compound library in Shionogi [36].

In this review, an overview of the developments and understanding of two of the proteases involved in very different points of the replication cycle of SARS-CoV-2 (Figure 3), the *viral* main protease (M^Pro^) and the *host* transmembrane protease serine 2 (TMPRSS2) is presented. On the one hand, viral protease M^Pro^ (previously known as 3-chymotrypsin-like protease) cleaves the new viral polyprotein produced in the early stages of the replication cycle of coronavirus at eleven conserved sites. It contains a Cys-His catalytic dyad and recognises and cleaves sequences such as a Gln–(Ser/Ala/Gly) [37]. On the other hand, host protease TMPRSS2 facilitates fusion between the virus and the host cell membrane. Specifically, the viral S glycoprotein is composed of two functional subunits: S1 that binds to ACE2, and S2, which contains several domains and is responsible for fusing the membranes of the virus and host [38]. The boundary between these two subunits, called S1/S2 cleavage site, is where proteolytic cleavage and activation take place by TMPRSS2, and this irreversible step is critical for the membranes’ fusion allowing the virus to penetrate the host cell [14,39,40,41].

## 2. Proteases as Targets

### 2.1. Proteases as Targets in Viral Infections

Proteases have been identified in DNA or RNA viruses as well as in enveloped or nonenveloped ones; they are endopeptidases that catalyze the cleavage of specific peptide bonds in viral polyprotein precursors or in cellular proteins [42]. These enzymes may use different catalytic mechanisms involving either Ser, Cys, or Asp residues to attack the cleavable peptide bond with a high degree of specific recognition and cleavage [42]. Proteases may possess a catalytic triad or dyad consisting of three or two amino acids at the active site that function together to perform covalent catalysis; a common motif could involve an acidic residue, a basic one, and a nucleophile [42].

Proteases are well-established targets for the treatment of viral infections, and those that have been more exploited in drug design are those of the hepatitis C virus, HIV-1, picornaviruses, herpes viruses, adenoviruses, and flaviviruses [43]. Considering their mechanism of action, these proteases can be inhibited by covalent inhibitors that will form a permanent covalent bond between a broken ester or amide functionality and some of the residues in the triad/dyad at the catalytic site. Additionally, protease inhibitors can work through a non-covalent mechanism competing with the natural peptide substrate [44].

Examples of protease inhibitors for hepatitis C virus (HCV) are boceprevir and simeprevir. Boceprevir is an oral protease inhibitor developed by Schering-Plough and Merck and approved by the FDA in 2011, and it was retired from the market in 2015 because of the enormous success of sofosbuvir as an HCV therapy [45]. Simeprevir was created by Medivir AB and Janssen and approved by the FDA in 2013; it is orally bioavailable and usually administered in combination with peg-interferon alfa and ribavirin. A specific inhibitor of HIV-1 protease is indinavir, which shows excellent bioavailability and was developed by Merck and approved for medical use in 1996, but its use is now discontinued [46]. Indinavir is the substrate and inhibitor for cytochrome P450 CYP3A4 isoenzyme resulting in common drug–drug interactions; for example, indinavir serum levels go up in the presence of antifungal azoles (CYP3A4 inhibitors) and down in the presence of rifampin or rifabutin (CYP3A4 inducers) [47]. Another protease inhibitor is nelfinavir, which is active against human immunodeficiency virus type 1 (HIV-1) but also shows good antiviral activity against herpes simplex virus 1 (HSV-1) and several other herpesviruses [48]. Additionally, some peptide-based compounds consisting of two to four natural or unnatural amino acids have been reported by Nitsche et al. to act as protease inhibitors of flavivirus and are also emerging as potent agents against dengue virus, Zika virus, and West Nile virus [49].

### 2.2. Viral M^Pro^ and Host TMPRSS2 Proteases in COVID-19: Overview of Reported Inhibitors

As previously mentioned, M^Pro^ and TMPRSS2 are two of the main proteases involved in the viral replication and entry of the virus into host cells, respectively. Hereafter, an overview of reported inhibitors of these proteases, which are summarised in Table 1, is presented.

**Table 1 pharmaceuticals-16-00834-t001:** Summary of the antiviral agents targeting MPro and TMPRSS2 proteases as therapies for COVID-19 described in this review, and their approval state.

Compound	Target	Company/Research Group	Approval State
Remdesevir	RNA polymerase (inhibitor)	Gilead	FDA and EMA approved, 2020
Molnupiravir	RNA polymerase (inhibitor)	Merck	FDA approved, 2021 Pending EMA approval
Paxlovid (Nirmatrelvir/Ritonavir)	Viral proteases, M^Pro^ and HIV protease (inhibitors)	Pfizer	FDA approved, 2021 EMA/UK/Canada approved, 2022
Ensitrelvir	Viral protease M^Pro^ (inhibitor)	Shionogy	Emergency approval Japan, 2022
Xiannuoxin (Simnotrelvir/Ritonavir)	Viral proteases, M^Pro^ and HIV protease (inhibitors)		approved in China in January 2023
Myricetin	Viral protease M^Pro^ (inhibitor)	[50]	Preclinical phase
Ebselen	Viral protease M^Pro^ (inhibitor)	[51]	Preclinical phase
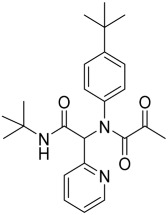	Viral protease M^Pro^ (inhibitor)	[52]	Preclinical phase
GC-376	Viral protease M^Pro^ (inhibitor)	Anivive Lifesciences	Preclinical phase
Boceprevir	Viral protease M^Pro^ (inhibitor)	[53]	Preclinical phase
Y180	Viral protease M^Pro^ (inhibitor)	[54]	Preclinical phase
MG-101,	Viral protease M^Pro^ (inhibitor)	[55]	Preclinical phase
Camostat	Host protease TMPRSS2 (inhibitor)	Ono Pharmaceutical	Approved for pancreatitis, Japan 1985. Phase 3 for COVID-19, 2021
Nafamostat	Host protease TMPRSS2 (inhibitor)	[56,57]	Approved as anticoagulant, Japan and Korea 2003 Phase 3 COVID-19, 2020
Gabexate	Host protease TMPRSS2 (inhibitor)	[58]	Preclinical phase
BC-11	Host protease TMPRSS2 (inhibitor)	[59]	Preclinical phase
Otamixaban	Host protease TMPRSS2 (inhibitor)	[60,61]	Preclinical phase
MI-432 and MI-1900	Host protease TMPRSS2 (inhibitor)	[62,63]	Preclinical phase
α_1_-antitrypsin	Host protease TMPRSS2 (inhibitor)	[64,65]	Preclinical phase

#### 2.2.1. Inhibitors of Viral M^Pro^

Even though several proteases have been identified in SASR-CoV-2, thus far, the main target of interest appears to be the main protease M^Pro^, which plays an important role in the replication cycle of the virus and is conserved among different variants of concern [66]. This viral protease contains a zwitterionic catalytic dyad, which is formed by Cys145 and His41. It has been proposed that the -SH group of Cys145 is the main player in the proteolytic activity, which is supported by His41. The catalytic process of M^Pro^ consists of the following steps: (i) His41 deprotonates the -SH group of Cys145; (ii) the thiolate anion nucleophillically attacks the C=O group of the amide in the substrate forming a tetrahedral intermediate; (iii) the peptide bond breaks forming the corresponding NH_2_ terminal group and recovering the neutral His41; (iv) the thioester formed in the nucleophilic attack is hydrolysed liberating the free COOH terminal group of the former peptide bond [67]. Therefore, this process can be inhibited by different mechanisms, in a covalent way by forming a covalent bond with Cys145 (i.e., ‘trapping’ Cys145) or in a competitive manner by occupying the active site of M^Pro^ not allowing access to the peptidic substrate.

This enzyme has no homologs in humans [68] offering an excellent target to deal with SARS-CoV-2 infections, and selective inhibitors have been proposed by structure-based drug design [69] or from natural compounds [70]; all the compounds here discussed are summarised in Table 1. Parallel to the development of paxlovid, the first M^Pro^ inhibitor approved in China in January 2023 has been xiannuoxin [71,72]. This is also a combination therapy of an M^Pro^ inhibitor (simnotrelvir, Figure 4) and ritonavir. Additionally, in terms of small molecule drugs, myricetin (Figure 4), which binds to Cys145 in its oxidised form, shows an inhibitory effect on M^Pro^ [50], and ebselen (Figure 4), which can establish selenium–sulphur (Se-S) interactions, also shows inhibition of this protease [51].

Many other non-peptide compounds have been found as inhibitors of the M^Pro^ of SARS-CoV with good activity, and they contain diverse cores such as pyrazolone [73], decahydroisoquinoline [74], asymmetrical aromatic disulphides [75], and octahydroisochromemes [76]. Hence, due to the large diversity of possible cores and their potential functionalization, these inhibitors of the M^Pro^ of SARS-CoV could not only be repurposed and tested as potential therapeutics for COVID-19 but also serve as springboards towards the development of novel SARS-CoV-2 antivirals. In general, their mechanism of action can be mainly classed as the non-covalent blocking of cysteine’s catalytic activity, as opposed to the covalent inhibition provided by peptidomimetic compounds, which is an attractive property in novel drug development.

Considering research already performed on the M^Pro^ of the SARS-CoV virus [77], it was possible to establish that this protease shares a 96% sequence homology with that of SARS-CoV-2 [78,79,80]. Both proteases exhibit four domains with the catalytic dyad located in a cleft between domains I and II and four subsites in the active site (S1′, S1, S2, and S4). Based on this similarity and the already existing research on inhibitors of the M^Pro^ of SARS-CoV, it is possible to consider drug repurposing for the new SARS-CoV-2 infection. 

Some promising novel small molecules had been discovered by Jacobs et al. identifying the diamide **1** (Figure 5) as a good candidate for the inhibition of the M^Pro^ of SARS-CoV [52]. From that starting system, they discovered a series of derivatives (Figure 5) with IC_50_ values in the low nanomolar range (<1 nM). Thus, they found that Ar^1^ and R groups play a minor role in drug activity, while large Ar^2^ groups such as fluorenes provided better activity. Additionally, different epimers were studied, and a clear trend of (R)-epimers having higher IC_50_ values than (S)-epimers was found. Among all the possible combinations of Ar^1^, Ar^2,^ and R, the best candidate was compound **2** (Figure 5) with an IC_50_ = 0.1 nM. Therefore, these systems could be promising ‘lead’ compounds to develop M^Pro^ inhibitors specific to the SARS-CoV-2 virus.

Furthermore, some peptidomimetic derivatives, especially those bearing an α-acyloxymethylketone warhead, have been reported to react with the catalytic Cys145 residue of the M^Pro^ of SARS-CoV-2 via nucleophilic addition, thus acting as a cysteine ‘trap’ [81]. For example, diamide GC-376 (developed by Anivive Lifesciences, Figure 6), which shows antiviral broad action against coronaviruses that produce feline infectious peritonitis, has also been found to inhibit SARS-CoV-2 M^Pro^, and its crystal structure complexed with M^Pro^ has been resolved (PDB:6WTT) [82]. In this crystal structure, it is possible to observe the formation of a covalent bond between an aldehyde intermediate (Figure 6) and Cys145. Additionally, the γ-lactam ring is accommodated in the S1 site forming hydrogen bonds (HBs) with His163, Glu166, and Phe140, the isobutyl group occupies S2, which is hydrophobic due to the residues that conform it (His41, Met49, and Met169), the carbamate functionality forms HBs with Glu166 and Gln189, and the adjacent phenyl methyl ester occupies the S4 pocket. Finally, the hemi-thioacetal covalently formed between GC-376 and M^Pro^ (Figure 6) is oriented towards the “oxyanion hole” formed by the peptide bonds connecting Gly143, Ser144, and Cys145.

As previously mentioned, the urea derivative boceprevir (Figure 6) is a protease inhibitor for the HCV virus, but it was shown to inhibit M^Pro^ in SARS-CoV-2, and the corresponding co-crystal structure was also recently resolved (PDB:6ZRU) [53]. In this structure, covalent inhibition is observed by the formation of a covalent bond between the C=O of the α-ketoamide moiety (Figure 6) and Cys145 forming a hemi-thioacetal similar to that seen with GC-376, occupying the aforementioned “oxyanion hole”. In addition, the cyclo-butylmethyl group occupies the S1 pocket, the dimethyl-3-aza bicycle moiety introduces into the hydrophobic S2 pocket, and the tert-butyl urea system gets into the S4 pocket (Figure 6).

Evidence for the mechanism of action of GC-376 comes from the elucidation of the stable hemi-thioacetal adduct via ^1^H NMR spectroscopy [83]. As shown in Figure 6, different modifications were introduced to improve the inhibitory power of these α-acyloxymethylketone derivatives, their metabolic stability, or their detectability; thus, fluorine (**3**, in Figure 6), deuterium (**4**, in Figure 6) [84], and indole (**5**, in Figure 6) derivatives were prepared with excellent results. From these outcomes, molecular complexity was eventually increased leading to the identification of drugs such as compound **6** in Figure 5 with a low nanomolar IC_50_. 

However, it is well known that peptide-like drugs have poor pharmacokinetic (PK) profiles; therefore, non-peptide small molecules would be preferred. Therefore, it is rather interesting to learn the work of Quan et al. who recently reported an orally available M^Pro^ inhibitor, Y180 (Figure 7), with an IC_50_ of 8.1 nM [54]. This compound, which contains an α-ketoamide, was prepared using the Ugi four-component reaction, and it is related to the non-covalent M^Pro^ inhibitors previously developed by Jacobs et al. (see Figure 6) [52]. Quan et al. have shown that Y180 protects against wild-type SARS-CoV-2, Alpha, Kappa, and Theta variants as well as against the omicron variant with excellent PK properties. The compounds designed by these authors exhibit an acetyl group attached to the carbonyl of an amide (in pink in Figure 7 right), thus allowing the formation of the so-called α-ketoamide warhead to interact in the S1′ pocket of the M^Pro^ active site. Additionally, to avoid epimeric conversion and keep the most active R configuration, they used deuterium at the chiral centre to avoid hydrogen exchange.

After resolving the crystal structure of the M^Pro^ complex with Y180 (PDB:7FAZ), they found that a covalent bond was formed between Cys145 in the M^Pro^ active site and the carbonyl C of the warhead transforming it into an alcohol. Besides the upstanding antiviral results obtained, Y180 does not show toxicity in vitro and in vivo, and even though the toxicity screen awaits confirmation in humans, this compound seems a very promising antiviral therapy for COVID-19.

Finally, it deserves to note the antiviral screening developed by Narayanan et al. utilising an in-cell protease assay, antiviral and biochemical activity evaluations, and structural assessments in order to quickly identify protease inhibitors as treatments for COVID-19 [55]. Thus, these authors used a library of 64 repurposed drugs, and by means of computational studies (i.e., docking), they analysed the suitability of catalogued compounds as inhibitors of two SARS-CoV-2 proteases M^Pro^ and PL^Pro^. In particular, they found three drugs, MG-101, lycorine HCl, and nelfinavir mesylate (Figure 8), as inhibitors of M^Pro^. However, when measuring in vitro inhibition of this protease, they found that only MG-101 was effective with an IC_50_ of 2.89 µM. These authors were able to elucidate the crystal structure of M^Pro^ with MG-101 (PDB:7LKD) identifying the formation of a covalent bond between Cys145 in the active site of the protease and the drug, thus indicating that this inhibitor blocks the binding of the natural substrate.

#### 2.2.2. Inhibitors of Host TMPRSS2

Much less is known about inhibitors of the other target of interest for COVID-19 therapies, the host protease TMPRSS2. In fact, this protease has only recently become a research hot topic due to its role in SARS-CoV-2 infection. TMPRSS2 consists of 492 amino acids and contains several domains: (i) an N-terminal intracellular region; (ii) a single-pass transmembrane domain; and (iii) a biologically active C-terminal extracellular region containing a binding site for calcium (i.e., LDLR-A receptor class), a scavenger receptor cysteine-rich (i.e., SRCR), and a serine protease region (SP) that cleaves at Arg or Lys residues in a sequence with a canonical Ser441-His296-Asp345 (UniProt numbering) catalytic triad (see PDB:7MEQ) [85,86]. TMPRSS2 is mostly expressed in the epithelial cells of tissues such as the prostate, ovary, breast, bile duct, kidney, pancreas, colon, small intestine, stomach, and lung. The most important residue in the active site of TMPRSS2 seems to be Ser441, which uses its hydroxyl group for the nucleophilic attack and trans-esterification of the corresponding peptide bond. In the activation of the SARS-CoV-2 spike protein S, the action of TMPRSS2 takes place at the junction of two sites, Arg685/Ser686 and Arg815/Ser816. Cleavage at Arg815/Ser816 results in the formation of the activated fragments (S1/S2 and S2′) needed for fusion and entry into the cell. Cleavage at Arg685/Ser686 occurs by establishing a HB and an ionic interaction between His296 and Arg682 as well as HBs and hydrophobic interactions between His296 and Ser441 with different residues around the cleavage site (Pro809, Lys814, and Ser810). This suggests that upon interaction with the S protein, TMPRSS2 experiences conformational changes that facilitate the alignment of the two cleavage sites of the S protein with the active site of the protease [62].

Most drugs targeting TMPRSS2 as antiviral COVID-19 agents are repurposed TMPRSS2 inhibitors that have already shown activity in other types of diseases. Thus, efficient inhibitors identified so far range from small organic molecules (i.e., camostat and nafamostat, Figure 9) [59,87] to proteins, such as aprotinin [62].

Camostat and nafamostat are 4-guanidinophenyl esters that act as covalent inhibitors undergoing trans-esterification in the TMPRSS2 active site, thus trapping the Ser441 involved in the catalytic triad (see mechanism of inhibition of camostat in Figure 10) [14]. Camostat was first reported as a mesylate salt in an antitumour study [88], and since then, it has also been shown to increase pancreatic secretions by increasing cholecystokinin release [89] as well as to inhibit serine proteases; its suitability has been investigated as antiviral for SARS-CoV-2 [14]. Camostat, developed by Ono Pharmaceutical, was approved in Japan in 1985 for the treatment of chronic pancreatitis and post-operative reflux esophagitis as well as to inhibit fibrosis in liver or kidney disease [90,91,92]. Its naphthalene analogue, nafamostat, has been used as an anticoagulant during haemodialysis in Japan and Korea for over 20 years [56], and it has also been reported to reduce spread intravascular coagulation in patients with haematological conditions [57]. Its serine proteases’ inhibitory activity is responsible for its use during haemodialysis by preventing the transformation of fibrinogen into fibrin as well as in pancreatitis by blocking clot formation [93]. A recent study proposes three mechanisms by which nafamostat is effective against COVID-19: an anti-coronavirus action by inhibiting TMPRSS2, anti-disseminated intravascular coagulation (anti-DIC) by inhibiting the fibrinolysis responsible of the cytokine storm observed in the disease and antiplasmin action, which interferes with the known pathogenicity of plasmin cleaving the viral S protein [94]. Even though camostat and nafamostat showed highly potent covalent inhibition of TMPRSS2, they have not been very successful in the clinical trials that are being carried out, probably because of their short half-lives in plasma [95]. A third TMPRSS2 and serine protease inhibitor that has been less mentioned in the literature and is also a guanidine derivative is gabexate (Figure 9), an investigational drug used as an anticoagulant and to decrease the production of inflammatory cytokines. Gabexate inhibits kallikrein, plasmin, and thrombin, components of the coagulation cascade, thus preventing the formation of fibrin, which is the culprit of clotting [58]. The inhibitory activity of gabexate versus TMPRSS2 was recently confirmed by Hall and co-workers to be IC_50_ = 130 nM [96]. In 2020, the same group proposed a computational model to explain the TMPRSS2 inhibition of gabexate, camostat, and nafamostat based on a homology model of the protease and considering a competitive mechanism of inhibition; however, when the crystal structure of the protease in a complex with nafamostat was resolved in 2021, this mechanism was proved to be wrong since nafamostat crystallised covalently bound to the protease product, indicating transesterification. Therefore, gabexate probably also works by the transesterification of the Ser441 in the active site of the protease [60].

Another interesting inhibitor of TMPRSS2 is the amidinium-containing small molecule BC-11, which was found by Günther’s group (Figure 10) [59]. This compound contains an unusual boronic acid moiety as well as carbamidothioate and seems to covalently inhibit protease activity by trapping the catalytic serine residue as camostat and nafamostat. In fact, the mechanism of action proposed for BC-11 is very similar to that of camostat or nafamostat involving the nucleophilic attack of the OH of Ser441 to the boron atom in the case of BC-11 (the C=O for camostat), forming a covalent bond to trap Ser441 (Figure 10). BC-11 shows a unique selectivity for serine proteases and, despite the modest inhibition showed for the SARS-CoV-2 omicron variant when given in combination with a spike glycoprotein inhibitor (i.e., AHN1-055 [97]), it shows a very good viral entry inhibition (e.g., at 25 µM each, a 50% inhibition of viral entry was achieved) making BC-11 an excellent lead compound. 

New inhibitors have been suggested following computational studies such as virtual screening; for example, otamixaban (Figure 11), which also contains an amidinium group, showed an IC_50_ = 0.62 μM against TMPRSS2 thus preventing viral entry [60]. Otamixaban is a clinical candidate that acts as an inhibitor of a serine protease involved in blood coagulation that transforms prothrombin into thrombin. In clinical trials, it has shown to be safe but not as efficient as the control used. It has reappeared again as a drug of interest because of its ability as a TMPRSS2 inhibitor and hence a potential COVID-19 therapy [61]. 

Additionally, amidino-containing peptidomimetics MI-432 and MI-1900 (Figure 11) were found to show activity against the spread of COVID-19 infection in vitro [63]. In a 2020 article, Bestle et al. showed that these TMPRSS2 inhibitors could also prevent SARS-CoV-2 infection and that this antiviral activity could be synergistically enhanced when combined with a furin inhibitor that acts at a different cleavage site [62]. Furthermore, another small peptidomimetic with nM potency was reported, N-0385 [98], showing a very high selectivity for SARS-CoV-2 in human lung cells by inhibiting cell entry of different variants of SARS-CoV-2 (i.e., Alpha, Beta, Gamma, and Delta). Compound N-0385 is a tripeptide mesylated in the N-terminal (i.e., CH_3_SO_2_-) and substituted by a ketobenzothiazol (i.e., kbt) in the C-terminal (i.e., Ms-Gln-Ph-Arg-kbt), and it seems to offer a high level of prophylactic and therapeutic benefit by blocking the activity of TMPRSS2 with an IC_50_ of 1.9 nM. Finally, it has been reported that the small peptide α_1_AT (α_1_-antitrypsin) inhibits TMPRSS2 protease activity in a dose-dependent manner, by leaving the S2 domain uncleaved and unable to undergo membrane fusion. The IC_50_ value obtained for α_1_AT as an inhibitor of TMPRSS2 in vitro was around 38.5 μΜ [64,65].

Finally, it is important to mention that even though the SARS-CoV-2 Omicron variant favours the endo/lysosomal cathepsin-L pathway over TMPRSS2 for cell entry, it has been recently shown that it still relies on TMPRSS2 for spreading into the respiratory tract, though not as much as other variants do [99]. Moreover, Bojkova et al. measured the effect of nafamostat on the replication of Omicron and Delta variants of SARS-CoV-2, and they found similar IC_50_ values for all variants (around 0.04 µM) [100]. Therefore, TMPRSS2 inhibitors are still important therapies for the treatment of SARS-CoV-2 infections.

## 3. Computational Studies and Modelling

Considering the difficult situation brought about by the COVID-19 pandemic and the pressure to find suitable therapies, an unprecedented amount of research initiatives was put in place. Bearing in mind the time required for the development of new drugs, emphasis was put on computational approaches that can facilitate predicting the suitability of already existing (i.e., repurposing drugs) or novel antivirals for the identified targets. Computer-assisted drug design strategies can be Structure-Based (when the 3D structure of the target is already known) or Ligand-Based (when the design is made by structural similarities with already known drugs). Since only a few drugs against coronavirus, in general, were known to be effective [101], the computational initiatives were focused on Structure-Based strategies, using well-known crystallographic structures of targets of interest, or employing developed models of those targets. Taking into account the huge amount of computational work produced since 2020, in this review, only a brief overview of a couple of examples using viral M^Pro^ and host TMPRSS2 proteases as targets will be presented.

### 3.1. Computational Studies with viral M^Pro^ as a Target

As mentioned, Structure-Based drug design requires knowing the 3D structure of the target; thus, in the case of M^Pro^, crystallographic studies show a dimer with two similar protomers (A and B) [102]. Each of these promoters contains the three domains already mentioned (I, II -antiparallel β-barrels- and III—mostly α-helices-) and the binding site with four sites (S1, S2, S4, and S1′) and the catalytic dyad (i.e., Cys145 and His41, between I and II), with the key residue Cys145 in the S1′ site [79]. Until today, 634 X-ray crystallographic structures of M^Pro^ have been stored in the Protein Database from 2003 (PDB:1P9S and 1P9U) until 2023 (PDB:7WQI, 7XAX, 7Z4S, 7Z2K, 7Z3U, 8IGN, and 8IGO, among others) [103]. Since many of these structures are in complex with small organic inhibitors or peptide substrates, they are very suitable templates for the design of novel inhibitors.

For example, in 2020, Jin et al. published the structure of SARS-CoV-2 M^Pro^ bound to a peptide-like inhibitor, N3 (Figure 12), which acts as a Michael acceptor (PDB:6LU7). This compound had been previously computationally designed by the same group as an M^Pro^ inhibitor for other coronaviruses such as SARS-CoV or MERS-CoV [104]. The crystal structure shows the presence of a water molecule interacting with His41 and another one within the active site interacting with Phe140, His163, and Glu166, stabilizing the “oxyanion hole” [67].

The crystal structure also resolves a covalent bond between the thiol group of Cys145 of protomer A and an alkene carbon of the N3 inhibitor (green box in Figure 12), thus confirming the Michael addition. The Gln-like system at the P1 substrate amino acid position of N3 (i.e., the lactam in the grey box in Figure 12) occupies the S1 subsite displacing the two water molecules. The isobutyl group at the P2 moiety inserts into the hydrophobic S2 subsite of protomer A, while the isopropyl at P3 is solvent-exposed, indicating that different functional groups could be introduced in this site. The methyl at the P4 substrate section lies inside a small hydrophobic pocket and the group in P5 establishes van der Waals interactions with Phe168. The benzyl group in P1′ also forms van der Waals interactions with several Thr residues in the S1’ subsite of protomer A [105]. The information provided by this crystal structure is highly important to find new inhibitors by means of in silico screening. Thus, Jin et al. carried out a virtual screening of their in-house library of compounds followed by docking of the most promising candidates. They found that cinanserin, a serotonin antagonist previously tested in humans, fits optimally in the M^Pro^ binding pocket, establishing cation-π interactions with His41 and Glu166. Afterward, they measured an IC_50_ = 125 µM for M^Pro^ indicating that, after further improvement, there is potential for this compound as a COVID-19 therapeutic.

Other examples of the application of computational tools in the development of M^Pro^ inhibitors are: (i) the already mentioned work of Oerlemans et al. who carried out molecular docking and resolved a co-crystal structure with boceprevir [53]; (ii) the thorough combination of computational tools used by Ngo et al. over a database of ~4600 compounds comprising virtual screening, fast pulling of ligand (FPL), and free energy perturbation (FEP), identifying darunavir as potential SARS-CoV-2 M^Pro^ inhibitor [105]; (iii) the approach followed by Semenov and Krivdin combining modelling of NMR chemical shifts and docking studies that found the natural compound berchemol to be a potential inhibitor [106]; (iv) the study of Yu et al. applying docking, molecular dynamics (MD) simulations and MM-GBSA methods with four HIV protease inhibitors and ribavirin which provided information on blocking M^Pro^ [107]; (v) Souza-Gomes et al. also used a combined approach using machine learning, docking, MM-PBSA calculations, and metadynamics with FDA approved compounds, and they found mirabegron to form the strongest interaction with M^Pro^ [108]; (vi) the study of Patel et al. who applied docking, MD simulations, the free energy of binding, and DFT calculations on a set of 7809 natural compounds and identified theaflavin and ginkgetin as M^Pro^ inhibitors [109]; (vii) the fluorinated tetraquinolines proposed by El Khoury et al. in a computationally driven study using high resolution MD free energy binding calculations and machine learning predictions [110]; or (viii) the 28 drugs proposed by Piplani et al. for repurposing as M^Pro^ inhibitors which resulted from docking studies followed by high-throughput MD simulations of large set of natural products and licensed drugs [111].

### 3.2. Computational Studies with Host TMPRSS2 as a Target

The 3D structure of the host protease TMPRSS2 complexed with nafamostat was not published until 2021 (PDB:7MEQ) [86]; therefore, when the pandemic was declared in 2020, modeling of potential TMPRSS2 inhibitors could only rely on homology models developed from related targets.

Probably the most used homology model until the crystal structure of TMPRSS2 was elucidated was that developed by Singh et al., where the full sequence of human TMPRSS2 (from UniProt database) was used and the protease domain was further built using the 3D structure of human plasma kallikrein co-crystallized with an inhibitor in the catalytic site (PDB:6O1G) as a template [112]. The root-mean-square deviation (RMSD) between the backbone atoms of the model and the experimental kallikrein structure was found to be 0.39 Å. It is well known that the catalytic site of TMPRSS2 is like that of plasma kallikrein, thus they identified this catalytic site as a major binding pocket. They also found an allosteric site with numerous aromatic residues which were solvent-exposed, referring to it as exosite. In summary, these authors produced homology models that were widely used for virtual screening studies. Another homology model widely used is that produced by Hussain et al. Thus, considering that hepsin shares high homological similarity with the S1 domain of TMPRSS2, they used the crystal structure of hepsin (PDB:1Z8G) as a template for that section and from there proceeded to develop a full model of TMPRSS2 using Modeller 9.16. The inter-residual distances between residues in the catalytical site of this model compare well to those observed in hepsin and plasma kallikrein [113].

All the studies performed with these homology models assumed competitive or allosteric non-covalent inhibition; however, when the crystallographic structure of TMPRSS2 in complex with nafamostat was published in 2021, it was clear that covalent inhibition was also a mechanism to consider for future design. Taking all this into account, some of the most recent computational work published is here discussed.

An interesting work that still makes use of a TMPRSS2 homology model is that of Manandhar et al. where they performed a virtual screening of 52,337 protease ligands (from the Zinc database) followed by MD simulations. Their virtual screen identified 13 hits, which were furtherly docked and followed up by the MD simulations resulting in only three hits that were able to establish a stable complex with TMPRSS2 by interacting with the Asp180, Gly184, Gly209, and His41 [114]. Sharma et al. carried out a computational study combining a virtual screen of a library of camostat-related compounds (from the PubChem database) and the reported crystal structure of TMPRSS2 (PDB:7MEQ), paired with molecular docking and MD simulations. They were able to identify seven compounds that were further docked, and their MM/PBSA free energy was calculated, giving four compounds with predicted improved potency over camostat [115]. The work carried out by Salleh examined the complex between TMPRSS2 and the S glycoprotein of SARS-CoV-2 also studying interactions of some TMPRSS2 polymorphic variants. They used docking to identify that some of the variants were ‘protective’ towards SARS-CoV-2 infections, whereas others were expected to increase susceptibility [116]. Bioinformatic and chemoinformatic (i.e., QSAR) methods were used by Serra et al. to prioritize potential therapies for COVID-19 through TMPRSS2 inhibition. The bioinformatic methods included dynamic dose-dependent MOA, connectivity mapping, and network-based drug targeting and provided a ranked list of potential inhibitors. QSAR methods were then employed, and the three best compounds were selected [117]. Finally, in a work disclosed but not yet peer-reviewed, Kondo et al. report the reaction energy profiles of the acylation of known TMPRSS2 inhibitors (camostat, nafamostat, and a nafamostat derivative) calculated by different QM methodologies considering solvation through the PCM approach. They conclude that the inhibitory activity of these compounds is related to the formation of a stable acyl intermediate, and hence the study of the reaction energy profiles in future derivatives could be used to predict TMPRSS2 inhibition [118].

## 4. Dual-Action Inhibitors

Finally, some published studies considering the potential inhibition of both viral M^Pro^ and host TMPRSS2 proteases by the same drug are discussed. Dual-action inhibitors involve the simultaneous blockage of two targets related to a specified disease, either by poly-pharmacology or by designing a single molecule to attack multiple targets. Thus, in the context of COVID-19 antivirals, a dual-action inhibitor combines two different active agents that synergistically reduce the viral infection. Some examples have been reported in the literature aiming at drugs that simultaneously inhibit the viral proteases M^Pro^ and PL^Pro^ [119], the viral protease M^Pro^ and the spike S protein [120], and, relevant to the present review, viral M^Pro^, and host TMPRSS2 proteases. Thus, Huang et al. investigated extracts of an herb used in traditional medicine (*Scutellaria barbata* D. Don) as a potential dual inhibitor of these two proteases. They measured the infection inhibition on cellular models (i.e., Calu3 and VeroE6 cells) combined with LC/MS analyses, and they found some extracts that inhibited SARS-CoV-2 infection in a TMPRSS2-dependent manner. Analysis of these extracts identified six compounds that were already known to inhibit either M^Pro^ or TMPRSS2, suggesting that this herb could prevent viral infection through a dual-action inhibition of both proteases [121]. In addition, in the line of natural products, Wang et al. screened a number of natural products from fruits identifying tannic acid as a potential dual-inhibitor for M^Pro^/TMPRSS2. The IC_50_ values measured for this compound were 13.4 μM for M^Pro^ and 2.31 μM for TMPRSS2, and functional assays showed that tannic acid blocks viral entry to the host [122]. Finally, in a thorough approach, Mahgoub et al. virtually screened more than two million molecules as TMPRSS2 inhibitor candidates, and the best compounds were then docked against M^Pro^ to find dual-target inhibitors. Then, they carried out MM-GBSA and predictive ADMET, and the QM-optimised structures of the most promising hits were found to optimally bind to both active sites. Further MD simulations indicated the potential of three compounds (i.e., Z751959696, Z751954014 and Z56784282) as dual-action inhibitors with adequate pharmacokinetic profile [123].

## 5. Conclusions

In this review, an overview of two proteases that play an important role in the infection by SARS-CoV-2 that is responsible for COVID-19 pandemic, the viral protease M^Pro^ and the host protease TMPRSS2, has been presented. Their different functions, structures, and inhibitors have been discussed, furtherly supported by the use of computational tools for atom-level understanding of their mechanism of action and design of new inhibitors. Finally, the potential of some compounds as dual-action inhibitors of both proteases has been discussed.

## Figures and Tables

**Figure 1 pharmaceuticals-16-00834-f001:**
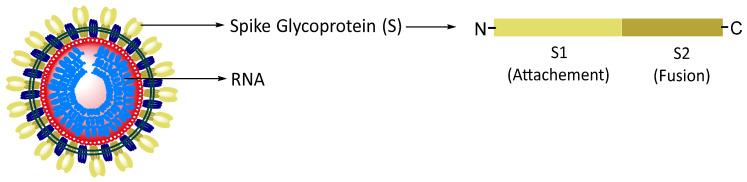
Scheme of the SARS-CoV-2 structure indicating the spike (S), its two subunits (S1 and S2), and large RNA genome.

**Figure 2 pharmaceuticals-16-00834-f002:**
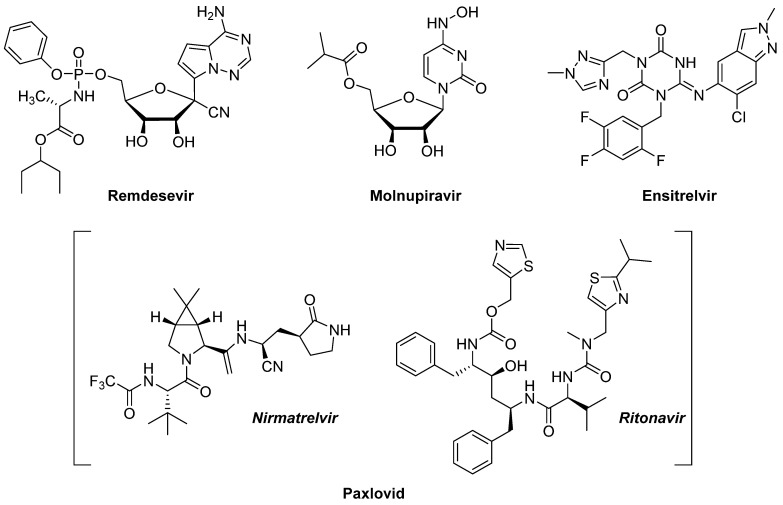
Structures of the three small-molecule drugs approved as therapeutic agents against SARS-CoV-2 infections: remdesivir, molnupiravir, ensitrelvir, and paxlovid (i.e., nirmatrelvir and ritonavir).

**Figure 3 pharmaceuticals-16-00834-f003:**
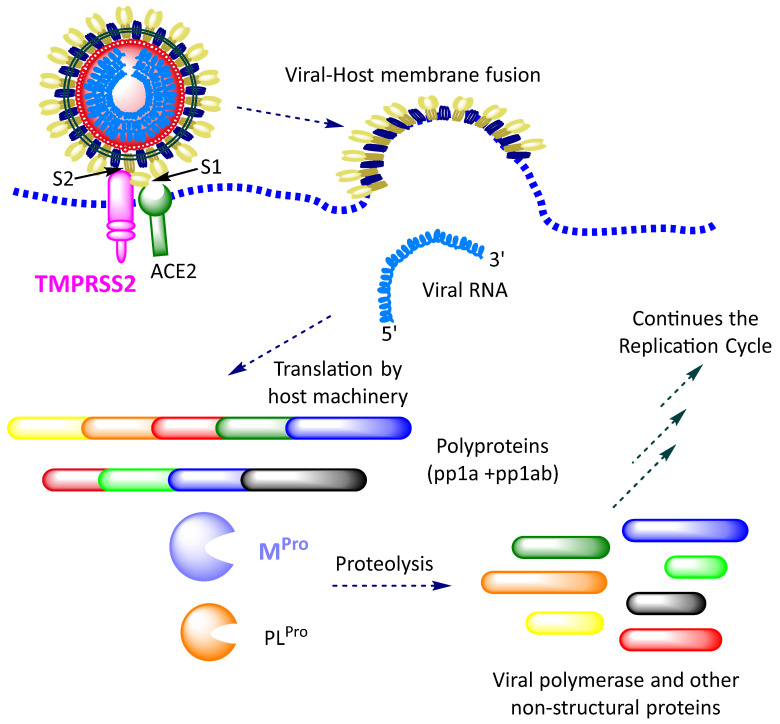
Schematic section of the replication cycle of SARS-COV-2 highlighting where MPro (mauve) and TMPRSS2 (pink) exert their actions.

**Figure 4 pharmaceuticals-16-00834-f004:**
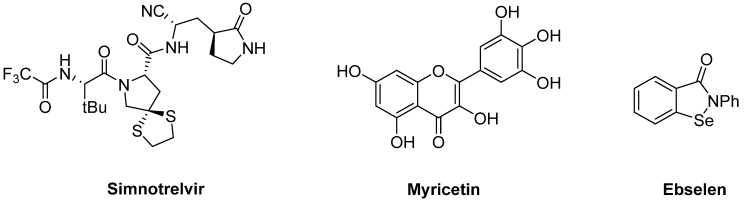
Structures of synthetic inhibitors of M^Pro^ simnotrelvir, myricetin, and ebselen.

**Figure 5 pharmaceuticals-16-00834-f005:**
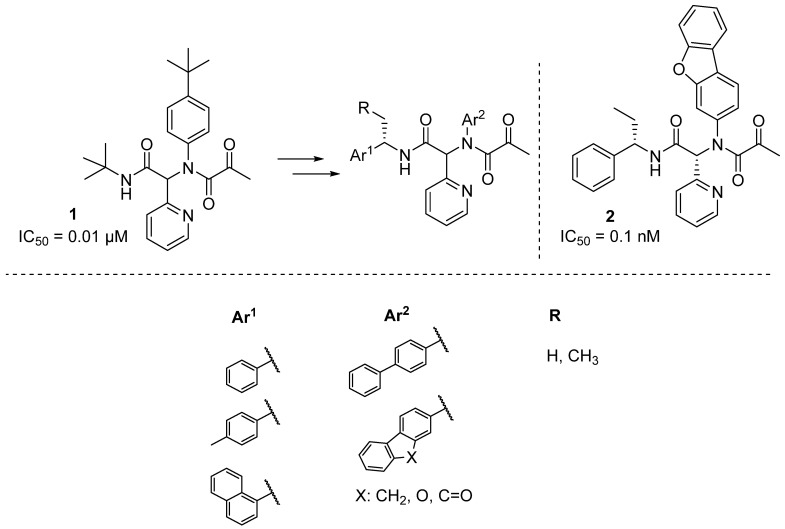
Structures of inhibitors of the M^Pro^ of SARS-CoV developed by Jacobs et al. [52].

**Figure 6 pharmaceuticals-16-00834-f006:**
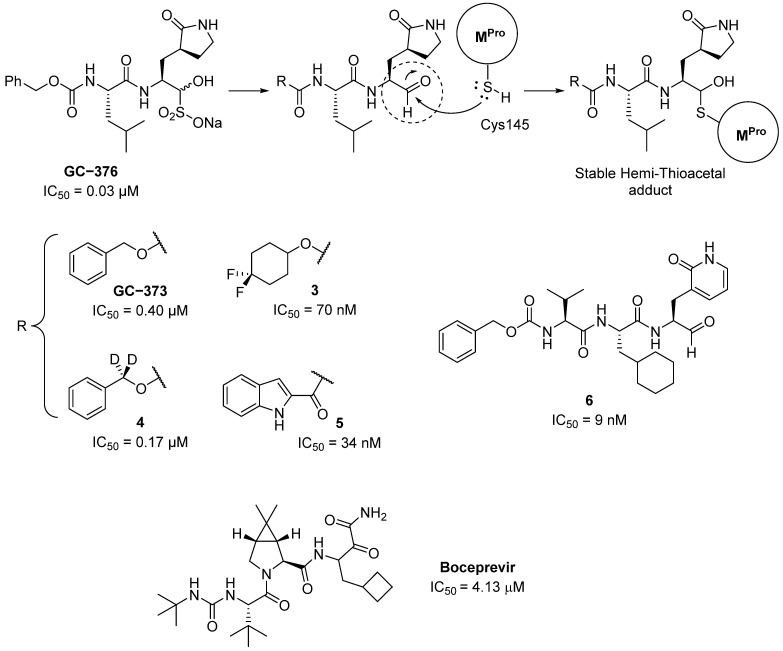
Peptidomimetics containing α-acyloxymethylketone that show activity as inhibitors of M^Pro^.

**Figure 7 pharmaceuticals-16-00834-f007:**
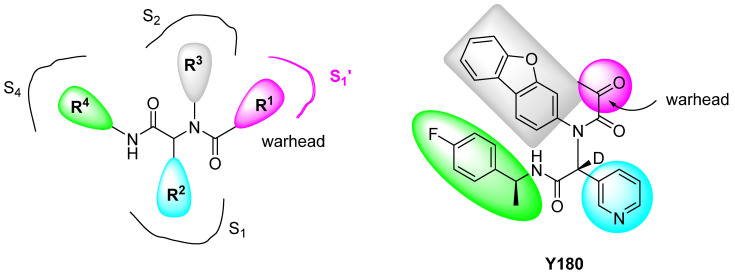
Diagram of the general structure of the α-ketamides prepared by Quan et al. [54] indicating the four elements (R^1^ to R^4^) required to interact with the different pockets in M^Pro^ binding site (**left**) and structure of the M^Pro^ inhibitor Y180 highlighting the interacting elements (**right**).

**Figure 8 pharmaceuticals-16-00834-f008:**
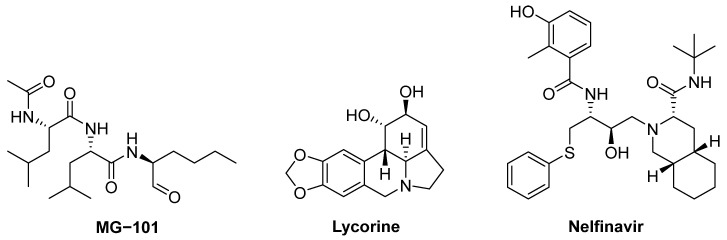
Structures of M^Pro^ inhibitors (MG-101, lycorine and nelfinavir) found by Narayanan et al. [55]. following a combined methodology from virtual screening to biochemical assays.

**Figure 9 pharmaceuticals-16-00834-f009:**
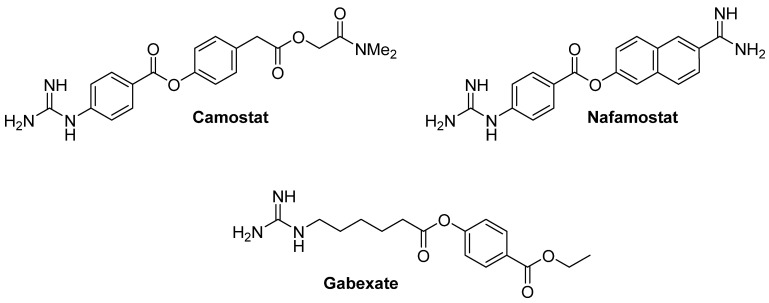
Guanidino-containing structures of TMPRSS2 inhibitors camostat, nafamostat, and gabexate.

**Figure 10 pharmaceuticals-16-00834-f010:**
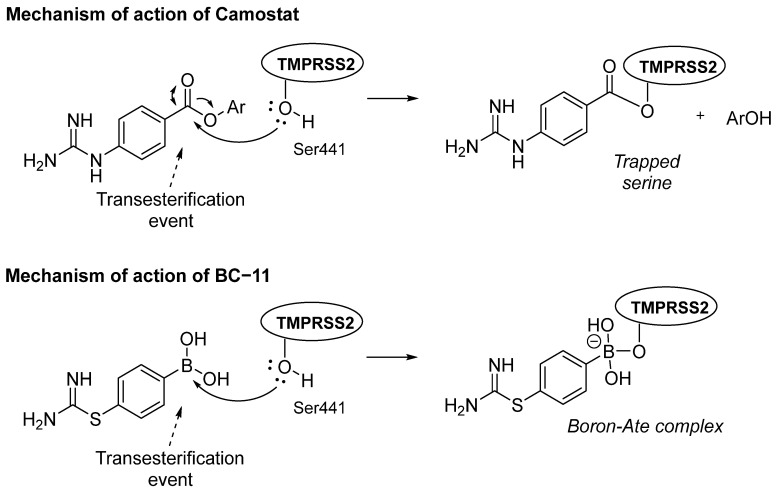
Schemes of the proposed mechanisms of inhibition of TMPRSS2 by camostat and BC-11.

**Figure 11 pharmaceuticals-16-00834-f011:**
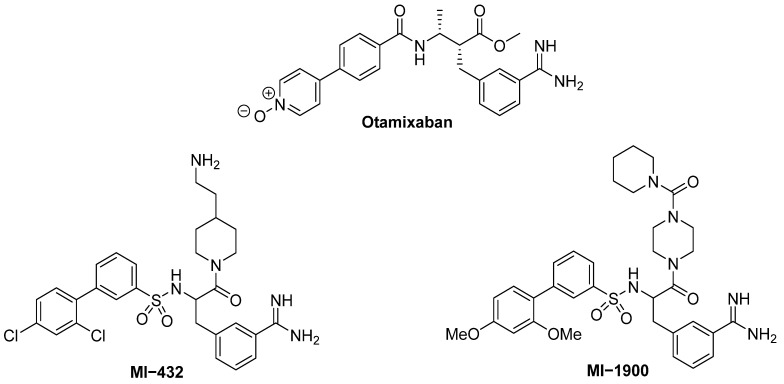
Amidino-containing structures of TMPRSS2 inhibitors otamixaban, MI-432 and MI-1900.

**Figure 12 pharmaceuticals-16-00834-f012:**
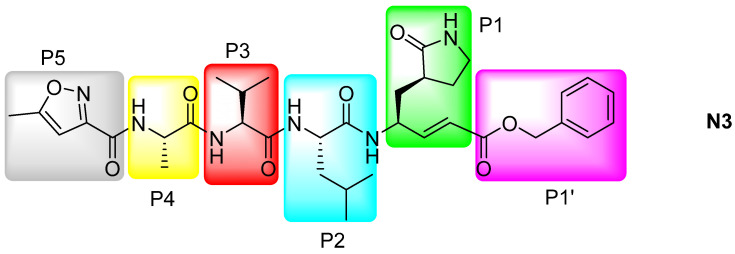
Structure of the computer designed MPro inhibitor peptide N3 indicating the different interaction moieties of the molecule P1–P5 and P1′.

## Data Availability

Data is contained within the article.

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
