# Peer review of "A Tale of Two Proteases: MPro and TMPRSS2 as Targets for COVID-19 Therapies"

_pharmaceuticals, 2023, doi:10.3390/ph16060834_

Round 1

Reviewer 1 Report

The review by Farkas et al, describes two interesting SARS-CoV-2 targets: the Mpro protease from the virus and the TMPRSS2 one from the host. They developed clearly in the review the pharmacological attempts done in the design and test of different small molecules to inhibit these proteases. They describe the mechanism of action of some of them and point out some of the computational efforts done in studying these two promising targets.  This review contributes significantly to the field, summarizing important approaches on this topic. My comments are that in part 4, the last part has a different letter format, so I recommend correcting it. Also, the references seem to have not a clear format; some of them have the DOI, others do not, and the names in the references have been vague format. 

Author Response

We thank the Reviewer for his/her comments. Here our response to the reviewer 1 comments:

R1-Comment-1: My comments are that in part 4, the last part has a different letter format, so I recommend correcting it.

Answer: This has been corrected.

R1-Comment-2: Also, the references seem to have not a clear format; some of them have the DOI, others do not, and the names in the references have been vague format. 

Answer: This has been corrected.

Reviewer 2 Report

This is a lengthy review, but weirdly has a few cited references (n=81). There are some serious problem with this manuscript; especially for review article, this issues should be addressed very well.

1. Claims made in the manuscript are not supported by sufficient references. Please cite the relevant reference on each sentence or claim (if the sentence contains two claims).

2. The review is lacking with visual presentation of S protein structure and how ACE2 and TMPRSS2 involved in the SARS-CoV-2 entry.

3.  The review is lacking with table presentation; I suggest you make two tables. One for FDA-approved  ACE2- or TMPRSS2-targeting drugs and another one for Non-FDA-approved. In the table, please show their respective efficacy (for FDA-approved you may search for the real-world or post-licensure data)

4. Please explain about the mutation in the S1 and S2 in SARS-CoV-2 variants.

5. Please include more studies that have unveiled the importance of  ACE2 and TMPRSS2 as drug targets such as 

1) Iqhrammullah et al.Sci. Pharm. 2023, 91(1), 15; https://doi.org/10.3390/scipharm91010015

2) Hasan et al., Narra J 2023. https://doi.org/10.52225/narra.v3i1.98

See more comments on the attached file.

English is easy to follow and no serious grammatical problems found in the manuscript.

Author Response

We thank the reviewer for his/her comments. Here our response to reviewer 2 comments:

General answer: We would like to clarify that the objectives of this Review are two proteases, one in the SARS-CoV-2 virus (MPro) and another in the host cell (TMPRSS2) as therapeutic targets to treat COVID-19. Other proteins or receptors such as the viral spike protein (S) or ACE2 are not the objective of this Review. Both the S protein and the ACE2 receptor are only discussed when talking about the viral replication cycle, but not as potential therapeutic targets. Accordingly, references suggested by Reviewer-2 that are not strictly related to MPro or TMPRSS2 as therapeutic targets will not be added to the revised manuscript since they will be out of the scope of the Review.

A comment about the number of cited references: looking at other reviews recently published in Pharmaceuticals (in 2023) we found that the number of references can vary between 32 and 172 hence, the 81 references in our Review seem like a reasonable number. In any case, following the advice of the Reviewer, we have added more references to the review up to a total of 112.

All figures have been drawn by the authors.

R2-Comment-1: Claims made in the manuscript are not supported by sufficient references. Please cite the relevant reference on each sentence or claim (if the sentence contains two claims).

Answer: We agree with the Reviewer that many statements in the manuscript were not fully supported with an appropriate citation and, following his/her advice we have now added all the references required.

R2-Comment-2: The review is lacking with visual presentation of S protein structure and how ACE2 and TMPRSS2 involved in the SARS-CoV-2 entry.

Answer: In Figure 1, a scheme indicating the two subunits of the S protein has been added and a diagram representing the points in the replication cycle of SARS-CoV-2 where MPro and TMPRSS2 exert their action has been included.

R2-Comment-3: The review is lacking with table presentation; I suggest you make two tables. One for FDA-approved  ACE2- or TMPRSS2-targeting drugs and another one for Non-FDA-approved. In the table, please show their respective efficacy (for FDA-approved you may search for the real-world or post-licensure data)

Answer: As mentioned, ACE2 is not covered in this review only the proteases MPro and TMPRSS2. In any case, as suggested by the Reviewer we have added a Table with the FDA and non-FDA approved drugs targeting MPro or TMPRSS2.

R2-Comment-4: Please explain about the mutation in the S1 and S2 in SARS-CoV-2 variants.

Answer: In the Review is mentioned that the Omicron variant seems to favour the endo/lysosomal cathepsin-L pathway over TMPRSS2 for cell entry, but that it has been recently shown that it still relies on TMPRSS2 for spreading into the respiratory tract, though not as much as other variants do. In addition, we mention the work of Bojkova et al. who found that the effect of nafamostat on the replication of Omicron and Delta variants of SARS-CoV-2 were similar. Considering the aims of this Review, details of the mutations of the S protein subunits are not relevant and are out of the scope of this Review. 

R2-Comment-5: Please include more studies that have unveiled the importance of ACE2 and TMPRSS2 as drug targets such as 

1) Iqhrammullah et al. Sci. Pharm. 2023, 91(1), 15; https://doi.org/10.3390/scipharm91010015

2) Hasan et al., Narra J 2023. https://doi.org/10.52225/narra.v3i1.98

Answer: As mentioned above neither the viral S protein nor ACE2 are the objective of the present review. Moreover, the review of Iqhrammullah et al. states that “Essential oils may prevent the SARS-CoV-2 infection by targeting its receptors on the cells (ACE2 and TMPRSS2). Menthol, 1,8-cineole, and camphor … serve as potential ACE2 blockers. beta-Caryophyllene may selectively target the SARS-CoV-2 spike protein and inhibit viral entry. … . In conclusion, essential oils could target proteins related to the SARS-CoV-2 entry and replication. Further studies with improved and uniform study designs should be carried out to optimize essential oils as COVID-19 therapies”; basically, no proof is presented indicating that any of these essential oils can really inhibit the enzymatic activity of MPro or TMPRSS2. Also, it is mentioned that “C. limon and P. graveolens could inhibit SARS-CoV-2 entry by downregulating ACE2 and TMPRSS2 mRNA expressions”; again, even if any of these essential oils really downregulate TMPRSS2 mRNA, they do not target the protease as potential inhibitors. Accordingly, this reference has not been included in the revised version of the manuscript.

The work of Hasan et al. deals with the S protein as a target for compounds extracted from ginger. Again, the S protein is none of the targets (proteases) which are covered in this review and for that reason this reference has not been included in the revised version of the manuscript.

R2-Comment-6: See more comments on the attached file.

Answer: All comments have been given due consideration.

Round 2

Reviewer 2 Report

I appreciate the authors for their careful revisions in the manuscript. The revised version is now can be considered to fulfill the publication criteria. Please pay attention on the writing errors which can be addressed during the proofreading stage. Last but not least, I would like to congratulate authors on their work. 

Please pay attention on the writing errors which can be addressed during the proofreading stage.